# Mortality and loss to follow-up among Tuberculosis patients on treatment in Meru County, Kenya: a retrospective cohort study

**Moses M. Ngari**[1,2]*, **Jane K. Mberia**[2], **Eunice Kanana**[3], **Deche Sanga**[4], **Martin K. Ngari**[5], **David N. Nyagah**[2], **Osman A. Abdullahi**[2]

1 KEMRI/Wellcome Trust Research Programme, Kilifi, Kenya, 2 Department of Public Health, School of Health & Human Sciences, Pwani University, Kilifi, Kenya, 3 Meru County TB Control Program, Meru, Kenya, 4 Kilifi County TB Control Program, Kilifi, Kenya, 5 Makueni County Referral Hospital, Wote, Kenya

* mngari@kemri-wellcome.org

## Abstract

Tuberculosis (TB) remains a leading cause of death globally. Patients who get lost to follow-up (LTFU) during TB treatment have high risk of relapse, mortality, treatment failure and developing Multidrug resistant TB. Empirical data to monitor long-term TB treatment outcomes in low-and-middle income countries (LMICs) are sparse. We determined proportion of TB patients who die or are LTFU during six months of treatment and identified factors independently associated with mortality or LTFU. A retrospective cohort using data from routine Meru County TB surveillance system was conducted. We included 38020 records of TB patients aged ≥15 years on treatment 2012–2022. TB treatment outcomes of interest were LTFU or death within six months of treatment. Survival analyses accounting for competing events were performed. Among the 38020 patients, 27608 (73%) were male and the median (IQR) age was 32 [25–42] years. 26599 (70%) had bacteriologically confirmed TB while 11421 (30%) were clinically diagnosed. During 16531 person-years of follow-up, 2385 (6.3%, 95%CI 6.0–6.5) and 1942 (5.1%, 95%CI 4.9–5.3) patients were LTFU and died respectively. In the multivariable model, patients on re-treatment after LTFU or after failure and those previously treated compared to new TB patients were positively associated with LTFU. Patients coinfected with HIV and those with unknown HIV status were positively associated with LTFU. In contrast, females, clinically diagnosed, extra-pulmonary TB and older patients were negatively associated with LTFU. HIV infected patients on ARVs, not on ARVs and unknown HIV status compared to not infected were positively associated with mortality. Other comorbidities (not HIV), clinically diagnosed, under-nourished and older patients were positively associated with mortality. More than 10% of TB patients either die or are LTFU before completing treatment. Targeted interventions are needed to improve treatment outcomes for TB patients who are at high risk of death or being LTFU.

**Data availability statement:** The study anonymized data and the analyses codes underlying the results presented are archived in the Harvard Dataverse repository: https://dataverse.harvard.edu/dataset.xhtml?persistentId=doi:10.7910/DVN/78JGUE.

**Funding:** The authors received no specific funding for this work.

**Competing interests:** The authors have declared that no competing interests exist.

## Introduction

Tuberculosis (TB) is a curable disease; however, it remains one of the leading causes of death globally [1–4]. In 2022, it was the second leading cause of death globally from a single infectious agent after COVID-19 [4]. Up to 11.4million people developed TB in 2022 in the whole world [4]. The United Nations Sustainable Development Goals (SDG-3) set an ambitious target to end the TB epidemic in 2030 by reducing the TB incidence by 80% and TB deaths by 90% from the 2015 levels. Despite the progress made in TB diagnosis through use of rapid culture/molecular techniques such as Xpert MTB/RIF assay, and clinical management, the global targets (SDG-3) are off-track [5]. In 2022, TB is estimated to have caused 1.3 million deaths, a reduction of 19% since 2015, which is below the End TB Strategy target of reducing TB deaths by 75% before 2025 [4,5]. Sub-Saharan Africa and South-East Asia continue to bear the highest burden of TB deaths, accounting for 81% of deaths in 2022 [4,6]. HIV which is endemic in Sub-Saharan Africa is one of the leading causes of deaths among people living with HIV (PLWHIV) and co-infected with TB despite the high uptake of ARVs [7–10]. Kenya is among the 30 high TB burden countries that contributed 87% of the global cases in 2022 [4]. Kenya reported 90560 drug-susceptible TB cases, a case notification rate of 168/100,000 population in 2022. In the same year, approximately 6% and 5% of drug-susceptible TB cases in Kenya died and were lost to follow-up respectively while 76% of new bacteriologically confirmed TB cases were cured [11].

Loss to follow-up (LTFU) during TB treatment remains another major threat to successful TB treatment. Patients who interrupt TB treatment have high risk of relapse, mortality, treatment failure and developing Multidrug resistant TB [12]. This means they remain infectious for longer period and therefore continue transmitting TB in the community [12]. Most studies usually report poor TB treatment outcome (combined treatment failure, death, loss to follow-up and not evaluated) and not LTFU separately. In China as high as 13% adult TB patients were LTFU from 2017 to 2021, while retreated patients and those with underlying medical conditions like hepatitis and cirrhosis were more likely to default treatment [13]. In Ethiopia, among children and adult patients, 5.36% and 16.6% were LTFU respectively while 13% were LTFU in urban informal settlement in Kenya [14–16]. Among multidrug resistant TB patients, up to 17% are LTFU before completing treatment [17]. However, in Brazil and Mozambique 18.1% and a high of 44.9% were LTFU respectively [18,19].

It is highly likely both mortality and LTFU during TB treatment will vary depending on the incidence of TB, prevalence of underlying medical conditions like HIV, incidence of multidrug resistant TB and access to health care. In Kenya, a few studies have examined mortality and LTFU during TB treatment [16,20–22], with some studies focusing on select population like paediatrics only [23]. Currently, there is limited empirical data to monitor long-term TB treatment outcomes in low-and-middle income countries (LMICs) and how the outcomes are impacted by evolving epidemiology of both infectious and noncommunicable diseases such as HIV, COVID-19, diabetes, hypertension and malnutrition. We aimed to determine proportion of drug-susceptible TB patients who die or are LTFU during six months of TB treatment in Meru County, Kenya and identify factors independently associated with mortality (adjusted for competing risk of LTFU) or LTFU (adjusted for competing risk of mortality).

## Materials and methods

### Study design

A retrospective cohort using data from routine standard National Leprosy and Tuberculosis and Lung Disease (NTLD) register.

## Study settings

The study analysed routine data from the TB electronic surveillance system from Meru County in Kenya. Meru County is located east of Mount Kenya with an estimated population of 1.5 million people in the 2019 census. Its population density in 2019 was 220 people per Sq.Km. The major economic activities are agriculture with Tea, Coffee, Khat, Bananas and Avocado as the top cash crops. The county is served by one regional referral hospital and approximately 176 public health facilities. The HIV prevalence in Meru County was 2.6% in 2020 and the ART coverage was 70% [24]. The county follows the Kenya national guidelines to diagnose and treat presumptive TB cases [25]. TB patients are diagnosed using World Health Organization (WHO)-recommended molecular TB diagnostics (Xpert MTB/RIF), smear microscopy and culture. Additionally, extra-pulmonary cases and those diagnosed based on clinical symptoms, suggestive histological examination or X-ray abnormalities without laboratory confirmation are clinically diagnosed and started on TB treatment. New TB patients are considered drug-susceptible and take Rifampicin (R), Isoniazid (H), Pyrazinamide (Z) and Ethambutol (E) during the intensive phase (first two months) followed by four months of treatment on Rifampicin (R) and Isoniazid (H) (2RHZE/4RH). However, new patients diagnosed with TB but with other complications such as meningitis and osteo-articular TB are treated with RHZE for the first two months followed by Rifampicin (R) and Isoniazid (H) for ten months (2RHZE/10RH). TB patients who previously had been on anti-TB including relapsed cases, were retreated in the first two months with Streptomycin (S), Ethambutol (E), Rifampicin (R), Isoniazid (H) and Pyrazinamide (Z), followed by Ethambutol (E), Rifampicin (R), Isoniazid (H) and Pyrazinamide (Z) for one month and Ethambutol (E), Rifampicin (R) and Isoniazid (H) for five months (2SRHZE/1RHZE/5RHE). The 2SRHZE/1RHZE/5RHE regimen was phased out in 2020 and replaced with 2RHZE/4RH. Patients diagnosed with drug-resistant TB are put on standardized regimen following WHO 2018 guidelines. At least 80% of drug-resistant TB patients are managed in the community where Community Health Promoters (CHP) offer directly observed Therapy (DOT) services. Each county maintains two separate databases for drug-susceptible and drug-resistant TB patients. All TB patients are systematically screened for malnutrition and offered HIV test at the time of starting TB treatment. Undernourished patients are offered nutritional support while HIV infected are referred to Comprehensive Care Center (CCC) which offers HIV management care. In all Kenya government health facilities, TB and HIV diagnosis and treatment are offered for free.

## Study population

We included records of drug-susceptible TB patients starting treatment within Meru County aged 15 years and above from January 2012 to December 2022 and registered in the County TB surveillance system.

## Data source and variables

Data used in this study were extracted from the Treatment Information from Basic Unit (TIBU) Meru County TB Electronic surveillance on 20th May 2024. Kenya transitioned from paper-based reporting to a national electronic surveillance system, i.e., TIBU in 2012. The system is operational in all Counties in Kenya that capture and store individual-level TB patient data which is reported to the Kenya national TB program. At each health facility, data are recorded into TB registers which are periodically transcribed electronically using tablets and synchronized to the national server [26]. Patients diagnosed with TB while in the wards are linked with outpatient TB clinics where they get registered in the TIBU system. The

study included the following variables collected in the routine data; demographic (age, sex), sub-counties, date of starting TB treatment, Clinical features including type of TB (pulmonary or extra-pulmonary), new or retreated patients, HIV status, other underlying medical conditions, TB diagnosis (either bacteriological or clinical), treatment regimen, Body Mass Index (BMI) a proxy for nutritional status, nutritional support provided and the type of direct observed treatment provided. A new patient was a presumptive TB case who has never been on TB treatment or had taken anti-TB medication for less than one month. TB patients on retreatment included; a) relapsed patients who were declared as completed treatment or cured but has been diagnosed with recurrent episode, b) treatment failures who included previously treated patients whose treatment outcome was declared as failure, c) treatment after loss to follow-up included patients previously treated but were categorized as loss to follow-up and d) other previously treated included those previously treated but outcome was unknown.

The outcomes of interest were the WHO TB treatment outcomes either death or loss to follow-up during six months of treatment. The deaths included TB patients who died for any reason on or after starting treatment but within the six months of follow-up. Loss to follow-up was defined to include TB patient who initiated TB treatment but experienced an interruption lasting at least two consecutive months [27]. Treatment success was defined as those who completed at least six months of TB treatment without evidence of treatment failure and those cured.

## Study size

Data from 38020 drug-susceptible TB patients on treatment from 2012 to 2022 within Meru County were included in the study. The study sample size (N = 38020) had a power of at least 90% to detect a hazard ratio ≥2.0 of HIV infected compared to HIV negative or clinically versus bacteriologically diagnosed TB being associated with mortality assuming 5% overall probability of death and two-tailed alpha of 0.05 [10,28].

## Statistical analysis

Data shared by the County TB coordinator were checked for completeness and processed for analysis. We found all records had a treatment outcome since we got data after at least one year of starting treatment (i.e., the last patient in the data started treatment on 31st December 2022 meaning they had completed treatment by 20th May 2024 when data were extracted). We assessed the pattern of missing data in the other variables and found low proportion (<10%) of missing data in HIV and BMI variables. The missing records were associated with the outcomes of interest and therefore were assumed not to be missing at random. Among the HIV and BMI variables, an extra category called `unknown' was added to represent the missing data and included in the analysis. The continuous BMI was categorized into: undernourished (BMI < 18.5), normal (BMI 18.5 to 25) and overweight (BMI ≥ 25). Age in years was categorized into groups of ten years apart from the open ended ≥65 years.

The data are reported as frequencies and proportions. Mortality and LTFU are reported as proportions with binomial exact 95% confidence interval. A non-parametric Wilcoxon-type test for trend was used to evaluate evidence of linear trend in annual proportion of deaths and LTFU from 2012 to 2022 [29]. Survival analysis was used to examine time to either death or LTFU after starting TB treatment and independent factors associated with the two outcomes separately. Person time was calculated from date of starting TB treatment up to date of the event or six months later (for those who completed treatment). Mortality and LTFU rates are reported as number of events per 1000 person-years and compared across groups using log-rank test. We tested proportional-hazards (PH) assumption using scaled

Schoenfeld residuals and found no evidence of PH violation for both outcomes. Time to event were plotted using Nelson-Aalen Cumulative hazard curves. To assess the effects of independent exposures on time to death or LTFU, two separate multilevel survival regression models were fixed for each outcome accounting for heterogeneity by sub-Counties. For the death outcome, we treated LTFU and transfer outside Meru County during the course of the six months as competing event because they precluded the probability of observing the event of interest (death). While for the LTFU outcome, deaths and transfer outside Meru County were treated as competing events. Therefore, the Fine and Gray competing risk method was used to examine independent factors associated with death or LTFU accounting for the sub-Counties heterogeneity and reported sub-distribution hazards ratios (SHR) as the measure of effects and their respective 95% confidence intervals [30]. We performed univariate analysis including only one independent variable and reported the crude SHR. To build multivariable models, we retained independent variables with P-value < 0.1 using backwards stepwise approach and reported adjusted SHR in the final model. All statistical analyses were conducted using STATA version 17.0 (StataCorp, College Station, TX, USA) and R (version 4.2.0).

## Ethical consideration

Ethical approval was granted by the Pwani University Ethics Review Committee (ISERC/PU-STAFF/003/2024) and permit to conduct the study granted by Department of Health, Meru County (MRU/GEN/GEN/C.50). Study participants and caregivers for children less than eighteen years old provided written informed consent for their data to be used. The analyses used anonymized data. The study is reported following STrengthening the Reporting of OBservation studies in Epidemiology (STROBE) [31] and REporting of studies COnducted using Observational Routinely-collected health Data (RECORD) [32].

## Results

### Patient characteristics

During the review period, 42116 TB patients were started on treatment of which 38020/42116 (90%) were ≥15 years old and included in this study. They included 27,608 (73%) male and their median (interquartile range - IQR) age was 32 [25–42] years. The patients were relatively young with over 50% younger than 35 years. The patients were more frequently (n = 29972, 79%) from a public health facility. Approximately one third (n = 12573, 33%) had normal BMI and 21331 (56%) were underweight. Bacteriologically confirmed TB cases were 26599 (70%) while 11421 (30%) were diagnosed based on clinical signs and abnormal Xray suggestive of TB. Among the 26599 bacteriologically confirmed cases, 10480 (39%) had both sputum smear positive and GeneXpert result, 14419 (54%) were sputum smear positive only while 1700 (6.4%) were GeneXpert positive only. A total of 33979 (89%) patients were new TB cases, 2333 (6.1%) were on retreatment after relapse, 851 (2.2%) were being retreated after LTFU (they were actively traced and restarted on treatment) and 652 (1.7%) were transferred into Meru County having started treatment in other Counties. There were 6773 (18%) TB patients co-infected with HIV while 1061 (2.8%) had unknown HIV status. Among the 6773 HIV infected patients, 6350 (94%) and 6698 (99%) were on ARVs and cotrimoxazole prophylaxis respectively. Overall, 280 (0.7%) had other comorbidities other than HIV. The most frequent comorbidities were:125 diabetes cases, 92 cases of hypertension and 29 cases of mental health conditions (S1 Fig). One thousand five hundred and ninety-four (4.2%) were on recreation drugs. The most frequent direct observed treatment was family-based (n = 35415, 93%). Thirty-six thousand and eleven (95%) were on the six-month first line treatment regimen of

2-month intensive phase of Rifampin (R), Isoniazid (H), Pyrazinamide (Z) and Ethambutol (E) followed by 4-month phase of Rifampin (R) and Isoniazid (H) (2RHZE/4RH) Table 1.

Clinical diagnosis, extra-pulmonary TB and other comorbidities apart from HIV were more frequent among the elderly (all P-values < 0.001) (S1 Table). HIV infection was more frequent among females, clinically diagnosed and extra-pulmonary TB cases (all P-values < 0.001) (S2 Table).

## Treatment outcome

A total of 32194/38020 (85%) successfully completed at least six months of TB treatment, and among those bacteriologically confirmed TB cases the cure rate was 13988/26599 (54%). There was no evidence of linear trend in annual proportion of patients who successfully completed treatment from 2012 to 2022 (Trend P-value = 0.70) Fig 1A. Two thousand three hundred and eighty-five (6.3%), 1942 (5.1%), 830 (2.2%), 342 (0.9%) and 327 (0.9%) were LTFU, died, transferred out of their treatment facility, failed treatment and developed drug-resistant TB and were moved to the drug-resistant cohort respectively (S2 Fig).

## Follow-up time and rates

The patients were on follow-up for 16531 person-years. During treatment, 2385 (6.3%, 95%CI 6.0 to 6.5) patients were lost to follow-up; incidence rate of 144 (95%CI 139–150) LTFU/1000 person-years. The median [IQR] time to LTFU was 68 [35–112] days. A total of 1084/2385 (45%) LTFU occurred within the first two months of intensive treatment, while cumulatively 1529/2385 (64%) LTFU occurred within three months of starting TB treatment. There was no evidence of linear trend in the annual LTFU rates from 2012 to 2022 (Trend P-value = 0.06) Fig 1B.

One thousand nine hundred and forty-two (5.1%, 95%CI 4.9 to 5.3) patients died: mortality rate of 117 (95%CI 112 to 123) deaths/1000 person-years. The median [IQR] time to death was 43 [18–88] days. A total of 1178/1942 (61%) deaths occurred within the first two months of intensive treatment, while cumulatively 1483/1942 (76%) deaths occurred within three months of starting TB treatment. There was no evidence of linear trend in the annual mortality rates from 2012 to 22 (Trend P-value = 0.11) Fig 1B.

LTFU and mortality rates per 1000 person-years are shown in Table 2. The rate of LTFU was highest in Igembe North Sub- County (196, 95%CI 177 to 218)/1000 person-years) and lowest in Meru Central Sub- County (53, 95%CI 42 to 67)/1000 person-years), Log-rank P-value < 0.001 (Fig 1C). Mortality rate was highest in Imenti North Sub- County (142, 95%CI 129 to 158)/1000 person-years) and lowest in Igembe North Sub- County (91, 95%CI 78 to 106)/1000 person-years), Log-rank P-value < 0.001 (Fig 1D). The rates of LTFU were not significantly different between patients with and without other comorbidity (Not HIV), on versus not on recreation drugs and across the DOT approaches (all log-rank test P-values > 0.05). Mortality rates were not significantly different between patients on versus not on recreation drugs and across the DOT approaches (all log-rank test P-values > 0.05) Table 2.

## Factors associated with LTFU or mortality

All the exposure variables explored for association with LTFU in the univariable analysis are shown in S3 Table. In the multivariable analysis, patients on re-treatment after LTFU (aSHR 3.28 (95%CI 2.59–4.17)) or after failure (aSHR 1.65 (95%CI 1.27–2.14)) and those previously treated (aSHR 2.11 (95%CI 1.66–2.68)) had higher hazard of LTFU compared to new TB cases (Fig 2A). Patients coinfected with HIV (on and not on ARVs) and those with unknown HIV status were positively associated with LTFU (Fig 2B). Compared to the

**Table 1. Patient characteristics at the time of starting TB treatment.**

| Characteristics | N = 38020 |
|---|---|
| Sex | |
| Male | 27608 (73) |
| Female | 10412 (27) |
| Age in years | |
| 15 to 24 | 9233 (24) |
| 25 to 34 | 11830 (31) |
| 35 to 44 | 8619 (23) |
| 45 to 54 | 4383 (12) |
| 55 to 64 | 2175 (5.7) |
| ≥65 | 1780 (4.7) |
| Year of starting TB treatment | |
| 2012 | 2860 (7.5) |
| 2013 | 3137 (8.3) |
| 2014 | 3337 (8.8) |
| 2015 | 3025 (8.0) |
| 2016 | 2939 (7.7) |
| 2017 | 3826 (10) |
| 2018 | 4250 (11) |
| 2019 | 3879 (10) |
| 2020 | 3382 (8.9) |
| 2021 | 3577 (9.4) |
| 2022 | 3808 (10) |
| Sub-County | |
| Buuri | 8295 (22) |
| Igembe North | 4245 (11) |
| Igembe South | 5596 (15) |
| Imenti North | 6030 (16) |
| Imenti South | 6496 (17) |
| Meru Central | 2761 (7.3) |
| Tigania East | 1606 (4.2) |
| Tigania West | 2991 (7.9) |
| Treatment facility type | |
| Public health facility | 29972 (79) |
| Private health facility | 7262 (19) |
| Prisons | 786 (2.1) |
| BMI group | |
| Undernourished (BMI < 18.5) | 21331 (56) |
| Normal (BMI 18.5 to 24.9) | 12573 (33) |
| Overweight (BMI ≥25) | 1529 (4.0) |
| Unknown/missing | 2587 (6.8) |
| TB diagnosis | |
| Bacteriologically confirmed | 26599 (70) |
| Clinical signs and X-ray | 11421 (30) |
| Patient category* | |
| New case | 33979 (89) |
| Re-treatment after relapse | 2333 (6.1) |
| Re-treatment after LTFU | 851 (2.2) |

*(Continued)*

**Table 1.** (Continued)

| Characteristics | N = 38020 |
|---|---|
| Transfer in | 652 (1.7) |
| Treatment after failure | 205 (0.5) |
| Type of TB | |
| Pulmonary TB | 33399 (88) |
| Extra-pulmonary TB | 4621 (12) |
| HIV status | |
| Negative | 30186 (79) |
| Infected | 6773 (18) |
| Unknown/missing | 1061 (2.8) |
| HIV treatment (ARVs) n = 6773 | |
| Started taking ARVs | 6350 (94) |
| Not yet | 423 (6.3) |
| On cotrimoxazole prophylaxis (n = 6773) | |
| Yes | 6698 (99) |
| No | 75 (1.0) |
| Other comorbidity* | 280 (0.7) |
| On recreation drugs$ | 1594 (4.2) |
| Direct observed treatment (DOT) | |
| Family-based | 35415 (93) |
| Community health Volunteer | 262 (0.7) |
| Healthcare worker | 2343 (6.2) |
| Treatment regimen | |
| 2RHZE/4RH | 36011 (95) |
| 2SRHZE/1RHZE/5RHE | 1350 (3.6) |
| 2RHZ/4RH | 304 (0.8) |
| RHZE/10RH | 195 (0.5) |
| Others | 160 (0.4) |
| Nutritional support | |
| No food support | 7268 (19) |
| Therapeutic/Supplementary food | 14924 (39) |
| Counselling only | 15828 (42) |

*;76 kidney disease, 52 hypertension, 18 neurodevelopmental disorders, 11 chronic obstructive pulmonary disease, 3 cancer, 3 asthma and 3 COVID-19, $;use of alcohol and smoking.

first-line treatment regimen, those on longer treatment period, i.e., 2SRHZE/1RHZE/5RHE had 34% higher hazard of being LTFU (aSHR 1.34 (95%CI 1.13–1.58)). In contrast, female patients were 36% less likely to be LTFU (aSHR 0.64 (95%CI 0.60–0.69)) and older patients had significantly lower hazard of LTFU relative to younger ones. Compared to patients from public health facilities, those from prisons had 38% lower hazard of LTFU (aSHR 0.62 (95%CI 0.47–0.82)). Clinically diagnosed patients and those with extra-pulmonary TB had lower hazard of LTFU. However, the BMI, other comorbidity (not HIV), on recreation drugs and type of DOT approach were not significantly associated with LTFU (Table 3).

Age had a linear trend association with mortality where the hazard of death increased from aSHR of 1.57 (25 to 34 years versus <25 years) to aSHR of 5.86 (≥65 years versus <25 years) test for trend P-value < 0.001 (Fig 2C). Compared to non-HIV infected, those infected

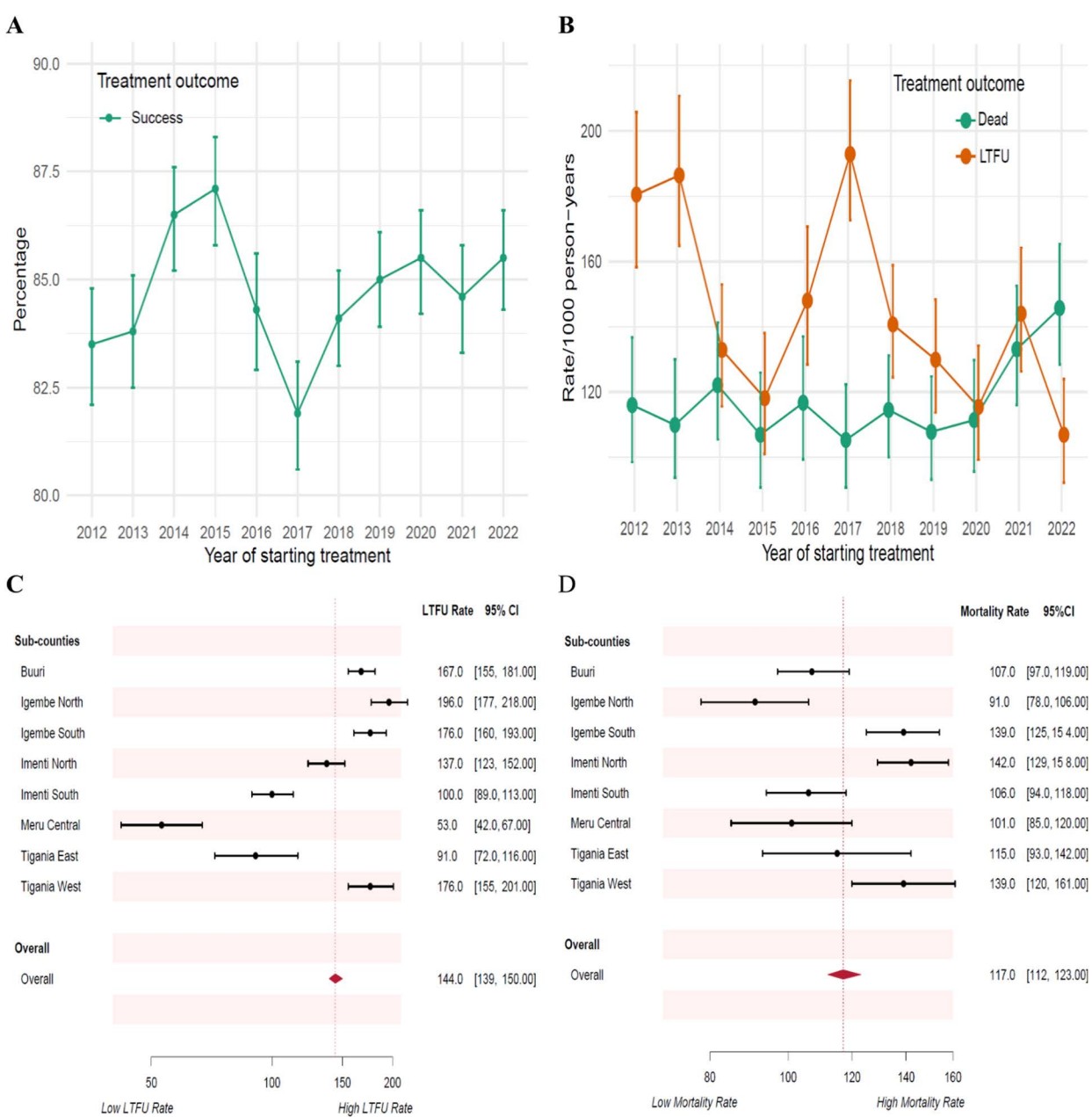

**Fig 1. Annual trend in: a) Successful TB treatment, b) Deaths and LTFU, c) Incidence rate of LTFU by sub-counties and d) Incidence rate of mortality by sub-counties.** Panel a represents annual percentage of successful TB treatment while b, c and d are LTFU or deaths rates per 1000 person-years.

on ARVs (aSHR 2.41 (95%CI 2.04–2.85)), infected and not on ARVs (aSHR 5.03 (95%CI 3.56–7.10)) and unknown HIV status were positively associated with mortality ([Fig 2D]). Other comorbidities (not HIV) were positively associated with mortality (aSHR 2.67 (95%CI 1.78–4.01)). Compared to bacteriologically diagnosed, clinically diagnosed patients had more than 2fold hazard of death (aSHR 2.22 (95%CI 1.97–2.49)). Undernourished patients had 50% higher hazard of deaths while the years 2021 and 2022 had significant higher mortality relative to 2012. However, patients from prisons had 30% lower hazard of death (aSHR 0.70 (95%CI

**Table 2. Losstofollow-up (LTFU) and mortality rates per 1000 person-years.**

| Characteristics | LTFU (N = 2385) | | | Deaths (N = 1942) | | |
|---|---|---|---|---|---|---|
| | N (%) | Rate/1000 person-years (95% CI) | Log-rank P-value | N (%) | Rate/1000 person-years (95% CI) | Log-rank P-value |
| Sex | | | | | | |
| Male | 1914 (6.9) | 160 (153-167) | <0.001 | 1334 (4.8) | 111 (106-118) | 0.0002 |
| Female | 471 (4.5) | 103 (94-113) | | 608 (5.8) | 133 (123–144) | |
| Age in years | | | | | | |
| 15 to 24 | 618 (6.7) | 151 (140-164) | <0.001 | 206 (2.2) | 50 (44–58) | <0.001 |
| 25 to 34 | 900 (7.6) | 175 (164-186) | | 466 (3.9) | 90 (83–99) | |
| 35 to 44 | 496 (5.8) | 132 (121–144) | | 462 (5.4) | 123 (112–135) | |
| 45 to 54 | 219 (5.0) | 116 (101–132) | | 315 (7.2) | 166 (149–186) | |
| 55 to 64 | 85 (3.9) | 93 (75-114) | | 219 (10) | 238 (209–272) | |
| ≥65 | 67 (3.8) | 92 (72-117) | | 274 (15) | 377 (335–424) | |
| Year of starting TB treatment | | | | | | |
| 2012 | 224 (7.8) | 180 (158-206) | <0.001 | 144 (5.0) | 116 (99–137) | 0.03 |
| 2013 | 256 (8.2) | 186 (165-211) | | 151 (4.8) | 110 (94–129) | |
| 2014 | 196 (5.9) | 133 (116-153) | | 180 (5.4) | 122 (106–141) | |
| 2015 | 158 (5.2) | 118 (101-138) | | 143 (4.7) | 107 (91–126) | |
| 2016 | 189 (6.4) | 148 (128-171) | | 149 (5.1) | 117 (99–137) | |
| 2017 | 315 (8.2) | 193 (173-215) | | 172 (4.5) | 105 (91–122) | |
| 2018 | 258 (6.1) | 141 (125-159) | | 210 (4.9) | 115 (100–131) | |
| 2019 | 218 (5.6) | 130 (114-148) | | 181 (4.7) | 108 (93–125) | |
| 2020 | 170 (5.0) | 115 (99-134) | | 164 (4.9) | 111 (96–130) | |
| 2021 | 224 (6.3) | 144 (126-164) | | 207 (5.8) | 133 (116–153) | |
| 2022 | 177 (4.7) | 107 (92-124) | | 241 (6.3) | 146 (128–165) | |
| Treatment facility type | | | | | | |
| Public health facility | 2000 (6.7) | 153 (147-160) | <0.001 | 1475 (4.9) | 113 (107–119) | 0.0001 |
| Private health facility | 348 (4.8) | 110 (99-122) | | 441 (6.1) | 140 (127–153) | |
| Prisons | 37 (4.7) | 114 (83-158) | | 26 (3.3) | 80 (55–118) | |
| BMI group | | | | | | |
| Undernourished | 1395 (6.5) | 151 (143-159) | <0.001 | 1152 (5.4) | 124 (117–132) | <0.001 |
| Normal | 688 (5.5) | 124 (115-134) | | 501 (3.9) | 91 (83–99) | |
| Overweight | 68 (4.5) | 101 (80-128) | | 69 (4.5) | 103 (81–130) | |
| Unknown/missing | 234 (9.1) | 218 (191-247) | | 220 (8.5) | 205 (179–234) | |
| TB diagnosis | | | | | | |
| Bacteriological | 1826 (6.9) | 156 (149-163) | <0.001 | 882 (3.3) | 75 (70–80) | <0.001 |
| Clinical signs | 559 (4.9) | 116 (107-126) | | 1060 (9.3) | 220 (207–234) | |
| Patient category | | | | | | |
| New case | 2025 (5.9) | 137 (131-143) | <0.001 | 1675 (4.9) | 113 (108–119) | <0.001 |
| Re-treatment after relapse | 122 (5.2) | 121 (101-145) | | 179 (7.7) | 178 (153–206) | |
| Re-treatment after LTFU | 172 (20) | 519 (447-602) | | 64 (7.5) | 193 (151–247) | |
| Transfer in | 42 (6.4) | 146 (108-197) | | 23 (3.5) | 80 (53–120) | |
| Treatment after failure | 24 (12) | 268 (180-400) | | 1 (0.5) | 11 (1.58–79) | |
| Type of TB | | | | | | |
| Pulmonary TB | 2208 (6.6) | 152 (145-158) | <0.001 | 1546 (4.6) | 106 (101–112) | <0.001 |
| Extra-pulmonary TB | 177 (3.8) | 90 (78-104) | | 396 (8.6) | 202 (183–222) | |
| HIV status | | | | | | |

*(Continued)*

**Table 2.** (Continued)

| Characteristics | LTFU (N = 2385) | | | Deaths (N = 1942) | | |
|---|---|---|---|---|---|---|
| | N (%) | Rate/1000 person-years (95% CI) | Log-rank P-value | N (%) | Rate/1000 person-years (95% CI) | Log-rank P-value |
| Negative | 1821 (6.0) | 136 (131–143) | <0.001 | 1118 (3.7) | 84 (79–89) | <0.001 |
| Infected | 449 (6.6) | 160 (146–175) | | 730 (11) | 260 (241–280) | |
| Unknown/missing | 115 (11) | 271 (226–326) | | J94 (8.9) | 222 (181–271) | |
| Other comorbidity (Not HIV) | | | | | | |
| No | 2376 (6.3) | 145 (139–151) | 0.07 | 1890 (5.0) | 115 (110–120) | <0.001 |
| Yes | 9 (3.2) | 81 (42–156) | | 52 (19) | 470 (358–617) | |
| On recreation drugs | | | | | | |
| No | 2282 (6.3) | 144 (138–150) | 0.73 | 1857 (5.1) | 117 (112–123) | 0.63 |
| Yes | 103 (6.5) | 152 (125–184) | | 85 (5.3) | 125 (101–155) | |
| Direct observed treatment (dot) | | | | | | |
| Family-based | 2210 (6.2) | 144 (138–150) | 0.69 | 1795 (5.1) | 117 (111–122) | 0.52 |
| Community health Volunteer | 17(6.5) | 153 (95–246) | | 14 (5.3) | 126 (74–212) | |
| Healthcare worker | 158 (6.7) | 151 (130–177) | | 133 (5.7) | 128 (108–151) | |
| Treatment regimen | | | | | | |
| 2RHZE/4RH | 2196 (6.1) | 140 (134–146) | <0.001 | 1800 (5.0) | 115 (110–120) | <0.001 |
| 2SRHZE/1RHZE/5RHE | 162 (12) | 283 (242–330) | | 103 (7.6) | 180 (148–218) | |
| 2RHZ/4RH | 12 (4.0) | 90 (51–158) | | 13 (4.3) | 97 (57–168) | |
| RHZE/10RH | 14 (7.2) | 171 (101–289) | | 20 (10) | 245 (158–379) | |
| Others | 1 (0.6) | 13.6 (1.91–96) | | 6 (3.8) | 81 (37–181) | |

All P-values are from log-rank test, the rates are per 1000 person-years.

0.55–0.90)) compared to those from public health facilities. Patient sex, type of TB, use of recreation drugs and type of DOT were not associated with mortality (Table 3).

## Discussion

In this large cohort of drug-susceptible TB patients on treatment, we found 5.1% died with no evidence of declining trend over the decade. Despite the improved understanding of the disease epidemiology and the investments in simplified diagnosis and free treatment in all health facilities in Kenya, there was an increase in TB Case Fatality Ratio during the last two years (2021 and 2022) relative to 2012 baseline which could partly be attributable to the disruptive effect of COVID-19 pandemic on health systems. The proportion of patients who died is within the range of Kenya national rate of 6% in 2022, and those reported in India, South Africa and previously in Kenya [20,33,34]. The first two months during the intensive treatment phase mark the most vulnerable period when almost two-thirds of deaths occurred. The high risk of early deaths could be a sign of poor health system preparedness which delays TB diagnosis and is unresponsive to needs of severely ill patients. In high HIV burden countries, patients co-infected with TB and HIV usually die very early following TB treatment [10,35,36]. However, as observed in this study, starting ARV treatment during the intensive treatment phase reduces mortality relative to being HIV infected but not on ARVs [37–39]. In fact, providing ARVs to patients with TB and HIV regardless of starting in the intensive treatment phase, reduces mortality by 44 to 71% [8]. Before 2016, in Kenya, HIV patients started ARVs when they have advanced HIV. However, following the WHO guidelines

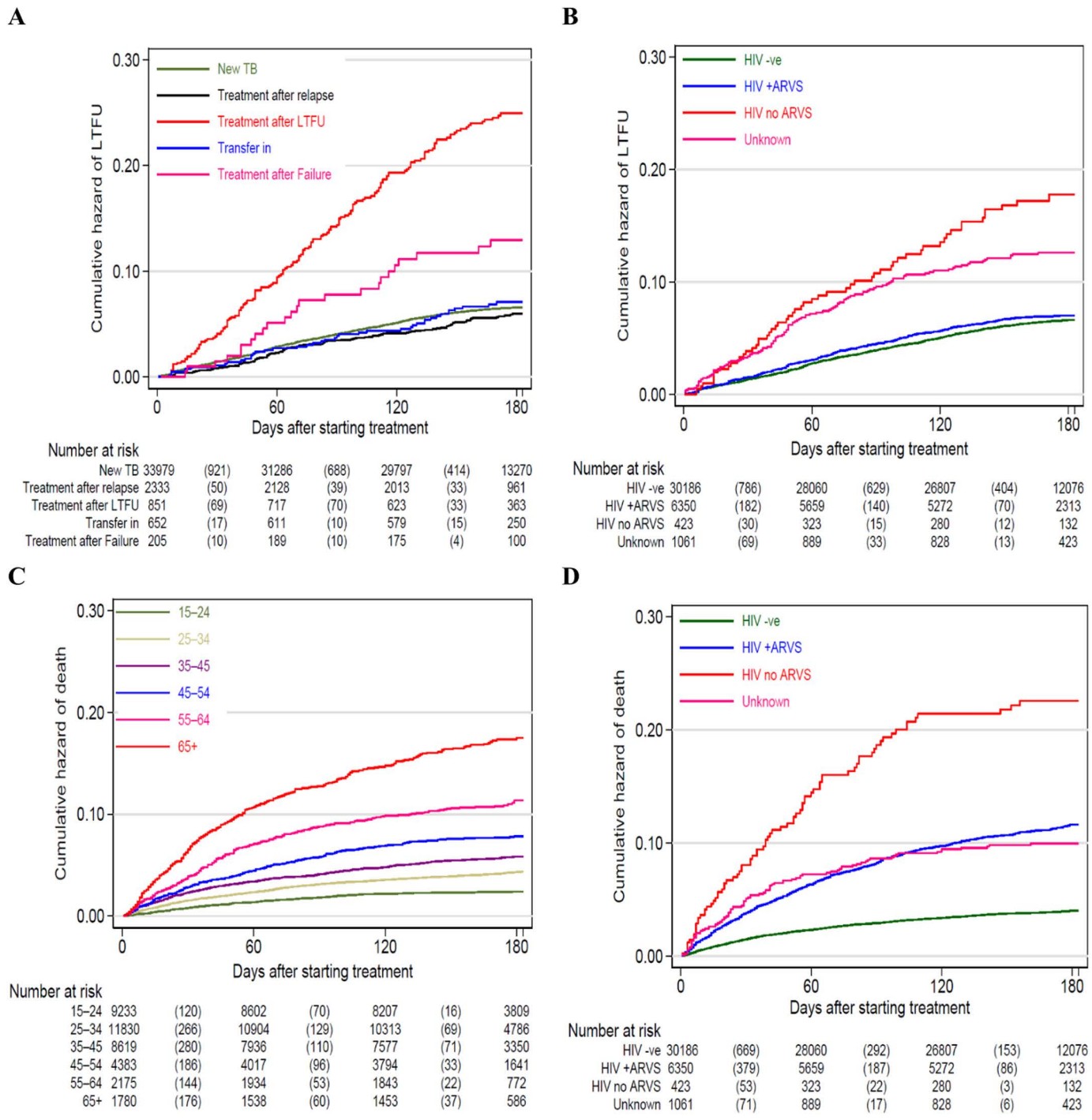

**Fig 2. Cumulative Hazard of LTFU by: a) Patient Type, b) HIV status and Cumulative Hazard for deaths by: c) age in years and d) HIV status.**

recommending immediate starting of ARVs upon HIV diagnosis in 2015, Kenya adopted the guidelines in mid-2016 [40]. Our results emphasize the need for implementing this policy to improve survival. But it is worth noting that ~3% of the patients had unknown HIV status despite the requirement of systematic screening for HIV among all TB cases for free in Kenya, and that those with unknown HIV status had high risk of mortality.

**Table 3.** **Multivariable analysis of patient characteristics associated with LTFU and mortality within six months of starting TB treatment.**

| Characteristics | LTFU | | Mortality | |
|---|---|---|---|---|
| | aSHR (95%CI) | P-value | aSHR (95%CI) | p-value |
| Sex | | | | |
| Male | Reference | | ¶ | |
| Female | 0.64 (0.60-0.69) | <0.001 | ¶ | |
| Age in years | | | | |
| 15 to 24 | Reference | | Reference | |
| 25 to 34 | 1.09 (0.96-1.23) | 0.18 | 1.57 (1.42-1.74) | <0.001 |
| 35 to 44 | 0.83 (0.73-0.95) | 0.008 | 1.89 (1.55-2.39) | <0.001 |
| 45 to 54 | 0.75 (0.62-0.90) | 0.002 | 2.40 (2.04-2.81) | <0.001 |
| 55 to 64 | 0.60 (0.44-0.82) | 0.001 | 3.48 (2.90-4.17) | <0.001 |
| ≥65 | 0.63 (0.51-0.78) | <0.001 | 5.86 (4.82-7.13) | <0.001 |
| Year of starting TB treatment | | | | |
| 2012 | Reference | | Reference | |
| 2013 | 1.06 (0.94-1.18) | 0.34 | 0.98 (0.77-1.26) | 0.91 |
| 2014 | 0.71 (0.51-1.00) | 0.05 | 1.13 (0.89-1.44) | 0.31 |
| 2015 | 0.67 (0.56-0.80) | <0.001 | 1.24 (0.97-1.59) | 0.08 |
| 2016 | 0.85 (0.60-1.19) | 0.34 | 1.42 (1.12-1.80) | 0.004 |
| 2017 | 1.10 (0.87-1.40) | 0.41 | 1.09 (0.93-1.29) | 0.29 |
| 2018 | 0.83 (0.74-0.93) | 0.001 | 1.15 (0.86-1.54) | 0.34 |
| 2019 | 0.76 (0.61-0.94) | 0.01 | 1.14 (0.90-1.46) | 0.29 |
| 2020 | 0.67 (0.55-0.81) | <0.001 | 1.20 (0.95-1.51) | 0.14 |
| 2021 | 0.81 (0.59-1.13) | 0.21 | 1.34 (1.05-1.70) | 0.02 |
| 2022 | 0.61 (0.48-0.77) | <0.001 | 1.40 (1.17-1.67) | <0.001 |
| Treatment facility type | | | | |
| Public health facility | Reference | | Reference | |
| Private health facility | 0.78 (0.60-1.00) | 0.05 | 0.85 (0.69-1.06) | 0.15 |
| Prisons | 0.62 (0.47-0.82) | 0.001 | 0.70 (0.55-0.90) | 0.006 |
| BMI group | | | | |
| Undernourished (BMI<18.5) | ¶ | | 1.50 (1.40-1.61) | <0.001 |
| Normal (BMI 18.5 to 24.9) | ¶ | | Reference | |
| Overweight (BMI ≥25) | ¶ | | 0.86 (0.76-0.98) | 0.03 |
| Unknown/missing | ¶ | | 2.06 (1.61-2.63) | <0.001 |
| TB diagnosis | | | | |
| Bacteriologically confirmed | Reference | | Reference | |
| Clinical signs and X-ray | 0.85 (0.73-0.99) | 0.03 | 2.22 (1.97-2.49) | <0.001 |
| Patient category | | | | |
| New case | Reference | | Reference | |
| Re-treatment after relapse | 0.77 (0.67-0.99) | 0.001 | 1.43 (1.12-1.82) | 0.004 |
| Re-treatment after LTFU | 3.28 (2.59-4.17) | <0.001 | 1.59 (0.93-2.73) | 0.09 |
| Transfer in | 1.06 (0.69-1.63) | 0.79 | 0.75 (0.40-1.39) | 0.36 |
| Previously treated | 2.11 (1.66-2.68) | <0.001 | 1.39 (0.77-2.49) | 0.27 |
| Treatment after failure | 1.65 (1.27-2.14) | <0.001 | 0.15 (0.02-1.19) | 0.07 |
| Type of TB | | | | |
| Pulmonary TB | Reference | | ¶ | |
| Extra-pulmonary TB | 0.72 (0.60-0.86) | <0.001 | ¶ | |
| HIV status | | | | |
| Negative | Reference | | Reference | |

*(Continued)*

**Table 3.** (Continued)

| Characteristics | LTFU | | Mortality | |
|---|---|---|---|---|
| | aSHR (95%CI) | P-value | aSHR (95%CI) | p-value |
| Infected on ARVs | 1.24 (1.07-1.44) | 0.003 | 2.41 (2.04-2.85) | <0.001 |
| Infected not on ARVs | 2.30 (1.90-3.03) | <0.001 | 5.03 (3.56-7.10) | <0.001 |
| Unknown/missing | 1.83 (1.26-2.65) | 0.001 | 2.27 (1.79-2.88) | <0.001 |
| Other comorbidity | ¶ | | 2.67 (1.78-4.01) | <0.001 |
| Treatment regimen | | | | |
| 2RHZE/4RH | Reference | | Reference | |
| 2SRHZE/1RHZE/5RHE | 1.34 (1.13-1.58) | 0.001 | 1.13 (0.86-1.50) | 0.37 |
| 2RHZ/4RH | 0.62 (0.33-1.18) | 0.15 | 0.91 (0.47-1.79) | 0.79 |
| RHZE/10RH | 1.54 (0.87-2.73) | 0.14 | 1.51 (1.06-2.15) | 0.02 |
| Others | 0.13 (0.59-0.30) | <0.001 | 0.50 (0.15-1.75) | 0.28 |

¶; independent variables not selected for inclusion in multivariable model, aSHR; adjusted Sub-distribution Hazard Ratio, aSHR are from multilevel competing risk analysis models.

Our results show patients who were elderly, undernourished, HIV infected including those with unknown HIV status, diagnosed using clinical signs, had other comorbidities and being treated after relapse had significantly high risk of death. These were not isolated findings because all these factors have been previously identified as risk factors for death among TB patients [2,41,42]. Advancing older age had a linear trend with increasing risk of mortality likely because of weaking immune responses among the elderly and age-related comorbidities which suggest the need for more support and care during TB treatment in this population [2]. As observed in this study and previously, the elderly more frequently had extra-pulmonary TB and clinical diagnosis which sometimes are hard to diagnose causing delays in starting TB treatment and thus increased mortality [41,43]. TB patients who are started on treatment based on clinical diagnosis usually will have a negative smear result and takes time to either diagnose extra-pulmonary or start treatment based on the clinical signs which can cause delay in treatment with a possibility of increasing mortality. Clinical diagnosis also has potential to under-diagnose other severe respiratory diseases and is more frequent among patients with HIV infection and the elderly [28,43]. The finding of patients on treatment after relapse had high risk of death suggests they were not cured which is not surprising given more than one-third of the patients were classified as `completed treatment' without evidence of negative smear test in the last two months of treatment. This could arise because of poor drug adherence, poor drug quality or treatment regimen inadequate for the population. Relapsed patients also include new infection likely caused by community high prevalence of infectious TB or exposure to different *M.tuberculosis* strain [44]. It is also likely the relapsed patient had higher risk of developing drug-resistant TB. The association with undernutrition was not surprising given it is the leading attributable risk factor for TB and malnourished patients have double risk of death from TB compared to non-malnourished patients [45,46]. Malnutrition impairs immune response and may alter how drugs are absorbed[45,47].

We observed a LTFU of ~6% within the six months of TB treatment which did not decline over the decade. It was within the range observed in Ghana (8.6%) [48], 5.3% in Kilifi, Kenya [22] and 6.8% in China[49]. However, it was lower than 13% in urban region in Nairobi, Kenya [16], 18.1% in Brazil [18] and 44.9% in Mozambique [19]. The differences could be due to varying HIV status, study settings and health system support availed to the patients during follow-up. Meru Central sub-county had the lowest LTFU, likely because of the support the

patients received from the community health promoters to collect their monthly drugs. During the intensive phase of TB treatment 45% of all LTFU occurred which was within the pooled range of 0% to 85% previously observed [50], highlighting the high risk of LTFU immediately after starting treatment. These patients who are LTFU present a challenge because they contribute to TB transmission in the community and acquisition of drug resistance [12].

Unlike the factors positively associated with mortality, elderly patients and those diagnosed using clinical signs had lower risk of LTFU. It is likely these group of patients which had very high risk of death felt adhering to TB treatment and the regular contacts with health workers would be of much benefit to their recovery. However, in Ethiopia, the risk of LTFU was higher among the elderly which could be attributed to varying social support systems available to the elderly in Kenya and Ethiopia [15]. TB patients who are on retreatment after previous treatment either after LTFU or treatment failure had remarkably high risk of LTFU. This highlights a sub-population of TB patients that would benefit from more social support such as the community-based DOT and involvement of family members during the treatment phase. As previously observed, HIV co-infected patients regardless of whether on ARTs or not had higher risk of LTFU which could be attributable to the huge burden of concurrent HIV and TB medication [22]. The introduction of immediate starting of ARVs upon HIV diagnosis has been associated with increased risk of LTFU in Uganda and Ethiopia, calling for a more cautious approach in implementing the policy [51,52]. It is likely the newly HIV diagnosed patients who also have TB are not adequately prepared for the lifelong ART medication in addition to the six months or more TB treatment or may not have severe HIV clinical signs thus likely to default treatment. Females were less likely to be LTFU because males who are predominantly household providers are more economically engaged and thus miss their follow-ups when working [53].

Our findings suggest deaths during TB treatment are predictable and preventable. Deaths mostly occur during the intensive phase of treatment and among the elderly with underlying comorbidities, advanced HIV infections and low BMI. A comprehensive assessment of the clinical needs for these patients during follow-up should be considered and risk stratified care offered. Future research to develop appropriate clinical support for example criteria for inpatient care referral and monitoring adherence to current guidelines such as systematic screening for HIV and malnutrition, immediate initiation of ARVs to those HIV infected and nutritional support should be prioritized [40,54]. We also identified LTFU as a significant challenge during TB treatment. A people-centred care recommended by the WHO in the End TB strategy has not been fully adopted in LMICs because of resource constraints [27,55,56]. Evidence of type of psychosocial support that works, and how they can be effectively implemented to reduce LTFU in LMICs are inconclusive [56,57]. A systematic drug susceptibility testing and appropriate treatment regimens for recurrent patients should be prioritized. A multi-sectoral approach is thus necessary to improve TB treatment outcomes.

## Study strengths and limitations

The strength of the study was the large number of patients systematically followed up during TB treatment and the availability of detailed TB treatment outcomes (the outcomes and actual dates of the outcomes). This was a passive surveillance using routinely collected data with its inherent limitations. Therefore, the analyses were limited to only available data. Data on adherence to treatment and social economical exposures were not collected. Data to identify patients referred for care from wards and those who got admitted to hospital during the six months of treatment were not available too. Data on reported reasons for TB treatment interruption were not available too. The study was only conducted in one County in Kenya and should be interpreted with caution.

## Conclusion

More than one tenth of TB patients either die or are LTFU before completing treatment. Our findings highlight the high burden of mortality and LTFU during the intensive phase of treatment. Interventions targeting TB patients on re-treatment, HIV infected, clinically diagnosed, undernourished and the elderly provides an opportunity to improve TB treatment outcomes.

## Supporting information

**S1 Fig. List of comorbidities reported other than HIV.**
(TIFF)

**S2 Fig. TB Treatment outcomes within six months of treatment.**
(TIFF)

**S1 Table. Participants characteristics stratified by age group.**
(DOCX)

**S2 Table. Participants characteristics stratified by HIV status.**
(DOCX)

**S3 Table. Univariate analysis of patient characteristics associated with LTFU and deaths within six months of starting TB treatment.**
(DOCX)

## Acknowledgement

We acknowledge study participants and Meru County TB program staff who managed and collected data from the patients. We thank Lilian Mwango for help with depositing data and analysis code to the data repository.

## Author contributions

**Conceptualization:** Moses M Ngari, Osman A. Abdullahi.

**Data curation:** Moses M Ngari, Jane K. Mberia, Eunice Kanana, Deche Sanga.

**Formal analysis:** Moses M Ngari.

**Investigation:** Jane K. Mberia, Eunice Kanana, Deche Sanga.

**Methodology:** Moses M Ngari, Jane K. Mberia, Eunice Kanana, Deche Sanga, Martin K. Ngari, David N. Nyagah, Osman A. Abdullahi.

**Project administration:** Moses M Ngari, Eunice Kanana, Martin K. Ngari, David N. Nyagah.

**Supervision:** Osman A. Abdullahi.

**Validation:** Osman A. Abdullahi.

**Visualization:** Moses M Ngari.

**Writing – original draft:** Moses M Ngari.

**Writing – review & editing:** Moses M Ngari, Jane K. Mberia, Eunice Kanana, Deche Sanga, Martin K. Ngari, David N. Nyagah, Osman A. Abdullahi.

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
