## [Decision Letter · Decision Letter 0]

17 Nov 2024

PGPH-D-24-02393

Mortality and lost to follow-up among Tuberculosis patients on treatment in Meru County, Kenya: a retrospective cohort study.

Dear Dr. Ngari,

Thank you for submitting your manuscript to PLOS Global Public Health. After careful consideration, we feel that it has merit but does not fully meet PLOS Global Public Health’s publication criteria as it currently stands. Therefore, we invite you to submit a revised version of the manuscript that addresses the points raised during the review process.

We look forward to receiving your revised manuscript.

Kind regards,

N. Sarita Shah

Academic Editor

Journal Requirements:

1. Please provide an Author Summary. This should appear in your manuscript between the Abstract (if applicable) and the Introduction, and should be 150–200 words long. The aim should be to make your findings accessible to a wide audience that includes both scientists and non-scientists. Sample summaries can be found on our website under Submission Guidelines: 

https://journals.plos.org/globalpublichealth/s/submission-guidelines#loc-parts-of-a-submission

2. We have noticed that you have uploaded Supporting Information files, but you have not included a list of legends. Please add a full list of legends for your Supporting Information files after the references list. 

Additional Editor Comments (if provided):

Reviewers' comments:

Reviewer's Responses to Questions

**Comments to the Author**

1. Does this manuscript meet PLOS Global Public Health’s publication criteria ? Is the manuscript technically sound, and do the data support the conclusions? The manuscript must describe methodologically and ethically rigorous research with conclusions that are appropriately drawn based on the data presented.

Reviewer #1: Yes

Reviewer #2: Yes

2. Has the statistical analysis been performed appropriately and rigorously?

Reviewer #1: Yes

Reviewer #2: I don't know

3. Have the authors made all data underlying the findings in their manuscript fully available (please refer to the Data Availability Statement at the start of the manuscript PDF file)?

Reviewer #1: Yes

Reviewer #2: Yes

4. Is the manuscript presented in an intelligible fashion and written in standard English?

Reviewer #1: Yes

Reviewer #2: Yes

5. Review Comments to the Author

Reviewer #1: This manuscript examines loss to follow up and mortality rates for patients with TB in Meru country over a 10-year period, 2012-2022. The authors highlight the high rate of loss to follow-up as well as the high mortality rates facing PLHIV, particularly during the early phase of TB treatment. Despite some ambiguities, the manuscript is well-written and the statistical methods are sound.

I provide below specific comments for the authors to address in the manuscript:

Title

• Line 1: Please correct “lost” to “loss.”

Abstract

• Line 26: Please edit “lost-to-follow-up” to “lost to follow-up.”

• Line 42. The sentence starting on line 42 is ambiguous. I suggest revising it to emphasize the high mortality among HIV/TB co-infected persons not on ARV.

• Line 46. Last sentence of the abstract. The sentence starting on line 46: “Targeted interventions are needed to improve treatment outcomes for TB patients who are at high risk of death or being LTFU.”

Introduction

• Line 63. Edit “bare” to “bear.”

• Line 69. Edit “Lost” to “Loss.”

• Line 83: I suggest editing “studied” to “examined” to avoid repetition.

Materials and Methods

• Line 99: Please spell out “Mt” to “Mount.”

• Line 131: Please edit “captures and stores” to “capture and store”

• Line 149: I suggest editing the sentence starting on this line to the following: “Loss to follow-up was defined to include TB patients who initiated TB treatment but experienced an interruption lasting at least two consecutive months. Treatment success was defined as those who completed…”

• Line 154: Please insert the word “sample” between study and size.

• Line 155: I suggest removing “presumptive” since the patients included in the study had received a confirmation of TB diagnosis either bacteriologically or clinically with chest X-ray findings consistent with TB.

• Line 157: Please edit “clinical” to “clinically”

Results

• Line 209: clarify ”more frequently” by stating the actual percentage of those who received care in a public health facility.

• Line 210: I suggest stating the percentage of those were underweight instead of overweight.

• Line 213-214: Correct “GenXpert” to “GeneXpert.”

• Line 224: Correct “Thousands” to “thousand” and “months” to “month.”

• Line 230: Correct “extra-Pulmonary” to “extra-pulmonary.”

• Line 236: Please include “%” after “5.1”

• Line 238: Please edit “resistant” to “drug-resistant TB.”

Discussion

• Line 301: Please correct “co-effected” to “co-infected.”

• Line 329: You should also consider the possibility of emerging drug resistance in patients receiving treatment after relapse as an explanation for the high mortality in patients on TB treatment after experiencing relapse. This would argue for drug susceptibility testing and appropriate treatment regimens for such patients.

• Line 343: The word “threat” is considered stigmatizing language in describing persons with TB. I suggest replacing it with “challenge.”

• Line 358: Please edit “new” to “newly.”

• Line 372: Please edit “resources contrains” to “resource constraints.”

• Line 374: I suggest rephrasing the last sentence to state: “A multi-sectoral approach is thus necessary for successful TB treatment.” or “A multi-sectoral approach is thus necessary to improve TB treatment outcomes.”

Tables:

• Table 1: Please edit the “$” sign from “On recreation drugs$”

• Table 1: Capitalize “dot” to “DOT”

• Table 2, Line 637: Please edit “Lost-to-follow-up” to “Loss to follow-up.”

Other comments

Please clarify whether drug resistance was available in Meru County. Do persons with drug-resistant TB have access to appropriate treatment in Meru County or do they have to travel to other locations?

It is unclear whether reasons for TB treatment interruptions are documented in the TIBU electronic system. Can the authors clarify whether some of the treatment interruptions could have been due to adverse drug events?

Reviewer #2: This is a well- written paper with appropriate analytical questions for LTFU and death to describe the TB response in Meru County, Kenya. The study addresses LTFU and mortality rates which are rarely studied.

Line 66: what is the TB case and mortality rate for Kenya for comparison to Meru findings

Line 246: Mortality rate instead of incidence rate?

Line234: is this a low cure rate?

Line 232: Was there a trend in re-treatment of LTFU to describe program efforts to track TB patients? What proportion of LTFU were not re-treated.

Line 277: Did an age/sex analysis show anything interesting? HIV analyses generally are conducted at this level for all clinical cascade elements to target program efforts.

Titles would be helpful for graphs.

What went right in Mery Central Sub-County?

6. PLOS authors have the option to publish the peer review history of their article (what does this mean? ). If published, this will include your full peer review and any attached files.

**Do you want your identity to be public for this peer review?** For information about this choice, including consent withdrawal, please see our Privacy Policy .

Reviewer #1: No

Reviewer #2: No

---

## [Editor Report · Decision Letter 1]

4 Dec 2024

Mortality and loss to follow-up among Tuberculosis patients on treatment in Meru County, Kenya: a retrospective cohort study.

PGPH-D-24-02393R1

Dear Dr. Ngari,

We are pleased to inform you that your manuscript 'Mortality and loss to follow-up among Tuberculosis patients on treatment in Meru County, Kenya: a retrospective cohort study.' has been provisionally accepted for publication in PLOS Global Public Health.

Best regards,

N. Sarita Shah

Academic Editor
